# Preliminary Analysis of Quantum Dots as a Marking Technique for *Ceratitis capitata*

**DOI:** 10.3390/insects16030270

**Published:** 2025-03-04

**Authors:** Richard Wimbush, Pia Addison, Francois Bekker, Minette Karsten, Melissa Powell, George Marais, Aaisha Moerat, Anandie Bierman, John S. Terblanche

**Affiliations:** 1Department of Conservation Ecology and Entomology, Stellenbosch University, Stellenbosch 7600, South Africa; wimbushd@sun.ac.za (R.W.); pia@sun.ac.za (P.A.); gulu@sun.ac.za (F.B.); minettek@sun.ac.za (M.K.); 22614583@sun.ac.za (M.P.); 23575417@sun.ac.za (G.M.); 23093323@sun.ac.za (A.M.); anandie@sun.ac.za (A.B.); 2Department of Conservation Ecology and Entomology, Centre for Invasion Biology, Stellenbosch University, Stellenbosch 7600, South Africa

**Keywords:** fluorescent powders, monitoring techniques, agricultural pest management

## Abstract

This study investigates the potential of quantum dots (QDs) as an alternative marking method for Mediterranean fruit flies (*Ceratitis capitata*) (Medflies) compared to traditional fluorescent powder marking. Medflies are significant agricultural pests that impact fruit crops worldwide, and effective marking techniques are essential for understanding, tracking, monitoring, and ultimately controlling their populations. Quantum dots are small light-emitting nanoparticles that have previously shown success in marking other organisms. In this study, the performance of QDs was tested against fluorescent powder in various aspects, including flight ability, recapture rates, and long-term marker stability. The results showed that while fluorescent powder consistently marked flies with high reliability, QDs faced challenges in marker retention and negatively affected flight performance. The findings suggest that while QDs have promise, further improvements are necessary for them to serve as an effective alternative to fluorescent powder methods currently employed in Medfly pest management.

## 1. Introduction

The South African agricultural industry accounts for 10% of the country’s total exports and provides an estimated 920,000 jobs [1,2]. Protecting crops from pests and diseases is crucial for this sector’s socioeconomic stability, necessitating sustainable pest management strategies. As the market increasingly shifts away from synthetic chemicals towards environmentally friendly methods [3], research into alternative control strategies is gaining urgency. Fruit flies (Family: *Tephritidae*) are among the most economically significant pests in South Africa, with the Mediterranean fruit fly (*Ceratitis capitata*), or Medfly, being one of the most destructive agricultural threats [4,5].

The Medfly is originally from sub-Saharan Africa, but has since spread to regions such as the Mediterranean, southern Europe, the Middle East, and parts of the Americas [6,7]. It is a polyphagous pest that affects a wide range of crops, feeding on more than 260 known hosts [8,9]. Female Medflies lay eggs beneath the fruit surface, with the emerging larvae causing internal damage, increasing the fruit’s susceptibility to secondary infections [10]. Their adaptability to diverse climates and ability to infest a wide range of crops make Medflies one of the most economically significant pests worldwide [11]. The presence of Medflies in exported fruit often leads to entire export shipments being rejected, forcing farmers to invest heavily in control measures to safeguard their crops [12].

The effective management of Medflies is essential to reducing economic losses. Various methods are available, including “attract and kill” bait traps [13], or traditional insecticides [14]. However, growing concerns about the associated environmental impact and emerging pesticide resistance have led to a market shift away from synthetic chemicals. One promising alternative is the Sterile Insect Technique (SIT), where sterile male Medflies are mass-released to control wild populations [15].

The Sterile Insect Technique is considered an effective and environmentally safe method for controlling Medflies, especially in smaller management zones [16,17]. This highly target-specific approach does not harm other species, making it preferable over chemical methods [18]. The Sterile Insect Technique’s success in eradicating fruit flies has been widely documented [15,19]. The development of genetic sexing strains further enhanced the SIT’s efficacy by allowing the rearing and release of only sterile males [20,21]. The SIT has been used in South Africa since 1997 when it was first introduced in the Hex River Valley to control Medflies over a 10,000-hectare area of table grapes [22]. The Sterile Insect Technique involves sterilized males being mass-released throughout the year, and out-competing wild males for mates, leading to a population decline, as the females lay infertile eggs [15]. The Sterile Insect Technique not only reduces the need for chemical pesticides but also helps boost agricultural yields by lowering fruit losses. Moreover, SIT programs have created job opportunities in the agricultural sector, contributing to economic growth [23].

The success of SIT control programs depends on the sensitivity of capture methods, the reliability of marking [24,25,26], and the dispersal of sterile fruit flies [27,28,29]. Marking the sterile males before release allows farmers to estimate the ratio of sterile to wild males later in traps, assisting in optimizing the number of sterile flies released for effective population control [17]. One common method of marking Medflies is the use of fluorescent powder dye, which is applied to Medfly pupae in a rotating drum, where the emerging flies are marked with dye in the ptilinum [30]. Under UV light, the marked sterile males are easily distinguished by the fluorescent glow compared to the wild types [31]. 

Insect marking is crucial in ecological research for estimating dispersal patterns, population dynamics, and behavioral activities [17,30,32,33,34]. It also helps in studies focusing on trophic relationships, territoriality, and other ecological interactions [35,36,37]. Additionally, the preservation and storage of marked insects are vital for long-term research, as they allow for the re-evaluation of specific individuals or population groups at a later stage. While fluorescent powder is widely used, it has several potential drawbacks. Some powder colors can be hard to differentiate (yellow–green combination) [38], and improper application may increase fly mortality and reduce mobility [39]. Studies have also indicated that fluorescent powders can transfer between flies during the insects’ death in traps or during same-sex interactions, compromising research accuracy [40]. The excessive use of fluorescent powders may block spiracles and reduce the flies’ lifespan [34]. Fluorescent powder has also been linked to increased mortality in other insects like *Dendroctonus frontalis*, *Ips grandicollis* beetles, and mosquitoes [24,39]. Additionally, tephritid grooming behavior may remove the marking, further reducing its effectiveness [17]. These limitations highlight the need for improved and alternative external marking techniques.

There is a clear need for an alternative external marker that is more user-friendly and easier to monitor compared to the conventional isotopes typically used for the internal marking of small insects [41]. One marking substance that could possibly be used as an alternative external marking technique is quantum dots (QDs). Quantum dots are microscopic semiconductor nanocrystals that emit visible and infrared light when exposed to UV light. This makes them highly versatile for various applications, including biological markers [41,42]. Traditionally, QDs have been extensively studied and utilized in the medical field for applications such as drug delivery, live imaging, and medical diagnosis [43,44]. Their ability to attach to virtually any biomolecule has also sparked interest in their use as marking agents for insects. Building on this, QDs have proven to be highly promising tools for both in vitro and in vivo labeling, thanks to their unique optical properties [45,46,47]. They have a broad excitation spectra and narrow emission bands which enable the fluorescence colors to be precisely adjusted just by modifying the composition and size of the QDs [48]. Additionally, when compared to traditional fluorescent powders, QDs offer superior photo-stability and emit a strong fluorescent signal [49]. These features make QDs easily detectable using handheld UV light, spectrophotometers, or fluorescence and confocal microscopes [49]. This also allows for the efficient marking of insect batches in various colors. Quantum dots have already been demonstrated to label and track pollen grains, highlighting their capability to mark even small items under field conditions [41]. Similarly, QDs were found to be effective external markers for Queensland fruit flies (*Bactrocera tryoni*) and diamondback moths (*Plutella xylostella*), both in laboratory conditions and during field recaptures [42]. Furthermore, QDs have successfully tracked the small aquatic organism *Daphnia magna* in 3D, using QD fluorescence with a multi-camera system, showing the broad applicability of QDs in tracking and marking studies [47]. Quantum dots have also demonstrated the ability to adhere to various parts of an insect’s body, including the head, wings, and legs [42], indicating their potential to be tailored for specific exoskeletal structures. This suggests that QDs could serve as promising external markers for insects, offering flexibility in their color range, minute size, and potential ability to be modified to suit a wide variety of insect exoskeletons/systems [42]. However, while QDs have shown promise in marking various organisms, challenges such as marker retention, as seen in *Tribolium castaneum* when exposed to abrasive materials like wheat, indicate the need for further refinement of these techniques [42]. Overall, the potential of QDs to advance mark–release–recapture studies in the field is significant, offering a modern alternative to traditional marking methods. 

For QDs to be successfully used in marking insects, they must meet specific criteria to ensure their effectiveness [17,49,50]. These criteria are outlined by having a straightforward application process that remains stable over time, as well as the ability to remain securely attached under various environmental conditions without interfering with the organism’s natural behavior. Additionally, QDs must be non-toxic, durable, and easily distinguishable from unmarked individuals. This study aimed to evaluate the efficacy of QDs as an alternative marking agent for Medflies compared to traditional fluorescent powders, with the goal of further extending the use of QDs within this tephritid species. The first objective was to determine the most efficient method for applying QDs to Medflies. Secondly, we assessed whether QDs impacted the Medflies’ flight ability, ensuring their natural movement remained unaffected. Thirdly, the recapture rates of QD-marked Medflies were compared with those marked using fluorescent powders in field assays to evaluate their suitability for mark–release–recapture programs. Additionally, under controlled laboratory conditions, we examined QD transferability between Medflies during interactions with unmarked flies to evaluate the risk of marking errors. Finally, the study assessed the persistence of QDs under different storage conditions to assess their long-term reliability.

## 2. Materials and Methods

### 2.1. Source of Ceratitis Capitata

The Mediterranean fruit fly pupae were obtained from Fruit Fly Africa, a mass rearing facility, located in Stellenbosch, South Africa. The larvae were maintained on an artificial diet of bran, yeast, sugar, water, and antimicrobial agents. Once the larvae were ready to pupate, they were placed in vermiculite for pupation and then subjected to gamma irradiation at 85 Gy. Upon arrival, they were either marked before incubation or directly incubated, depending on the experiment. Incubation was carried out at 25 °C and a photoperiod of 12:12 L:D. After emergence, the adults were given sugar and water ad libitum separately.

### 2.2. Fluorescent Powders

The three fluorescent powder colors (yellow, blue, and pink) were obtained from DayGlo^®^, Color Corp (DayGlo^®^ Color Corp., Cleveland, OH, USA). All experiments used just the fluorescent powder alone (without a solvent) at a concentration of 40 mg of fluorescent powder per 1000 Medfly pupae, except in the “suitable marking technique” experiment. Pink fluorescent powder was applied at a concentration of 2 g per liter of pupae [51]. A volumetric measurement of 1000 pupae yielded 20 mL, allowing for the calculation of a final application ratio of 0.04 g of powder per 1000 pupae. Fluorescent powder was added directly to the Petri dish containing Medfly pupae and gently rotated and shaken for 30 s to ensure an even distribution across the pupae. To observe the fluorescent powder marking results, the marked adults were examined under a stereomicroscope equipped with UV light (Appendix A, Figure A1).

### 2.3. Quantum Dots

The QDs used in this experiment were obtained from Strem Chemicals, supplied in 50 mg glass vials (Strem Chemicals, Inc., Newburyport, MA, USA).The three QD colors used in this study were all Copper Indium Disulfide/Zinc Sulfide QDs, with a particle size of 5–10 nm and a quantum yield exceeding 75%. The specific characteristics of the three colors—yellow, orange, and red—are as follows: yellow QDs had a peak emission of 550 nm ± 10 nm and a Full Width at Half Maximum (FWHM) of 115 nm ± 20 nm; orange QDs had a peak emission of 590 nm ± 10 nm and an FWHM of 120 nm ± 20 nm; and red QDs had a peak emission of 630 nm ± 10 nm with an FWHM of 125 nm ± 20 nm. The QDs were suspended in 99% ethanol at a concentration of 7.5 mg/mL and subsequently applied at a concentration of 160 µL QD solution per 1000 Medfly pupae, except in the “suitable marking technique”. This equates to 1.2mg of QDs per 1000 Medfly. The QDs needed to be suspended in a solvent in order to develop a novel straightforward and fast marking method. Ethanol was chosen due to its volatility, allowing it to evaporate quickly and leave the QDs deposited on the fly pupae/adult. A limitation of applying QDs in ethanol was that they did not dissolve uniformly within the solution. Therefore, prior to application, the solution was spun briefly in a standard benchtop centrifuge to allow the QDs to settle at the bottom of the Eppendorf tube. Before application, the solution was gently agitated to disperse the QDs evenly into the solution, after which they were pipetted onto a Petri dish containing the pupae or adults, depending on the experiment. The Petri dish was then lightly shaken to evenly distribute the QDs, and the lid was left off to allow the ethanol to evaporate quickly. To observe the QD marking results, the marked adults were examined under a stereomicroscope equipped with UV light (Figure A2 and Figure A3). 

### 2.4. Suitable Marking Technique

This experiment examined the efficacy of QDs as markers on Medflies compared to fluorescent powder using a novel QD marking technique. This was tested in three ways: Medfly pupae marked with QDs (43 μL per 50 pupae), Medfly adults marked with QDs (43 μL per 50 pupae), and Medfly pupae marked with fluorescent powders (0.0008 g per 50 pupae). Male and female flies were kept separately and maintained in 1 L ventilated containers, incubated for three days, and left for an additional five days to reach maturity. Flies were then chilled at 5 °C and screened for marked individuals using a stereomicroscope and UV light, with the percentage of marked individuals recorded. Due to the small sample size used in the experiment, the concentration of fluorescent powder was adjusted to prevent over-marking, and the quantity of QDs was increased to help visualize them better. This adjustment was necessary because working with extremely small quantities was not feasible with the available equipment, and the original concentrations were designed for large-scale applications. However, as the objective of the experiment was to determine the most suitable marking technique rather than optimize specific quantities, these adjustments did not significantly impact the overall findings. Each replicate was repeated four times, with each replicate comprising three treatments, each containing 20 specimens (N = 240).

### 2.5. Laboratory Flight Ability

The flight trial was conducted in a controlled environment to assess Medfly flight ability under the different marking treatments. Sixty adult Medflies were divided into three groups of twenty; the two marking techniques (fluorescent powder or QDs) and an unmarked control group. Each group was tested separately to prevent cross-contamination. The Medflies were placed on a stage surrounded by water in a 60 × 60 cm mesh cage equipped with a light source and a feeding platform near the roof. The feeding platform was stocked with yeast extract, water, sugar, and guava juice to encourage flight activity. Removable sticky traps were positioned around the feeding platform to capture flying flies, while the water barrier around the stage prevented flies from walking towards the food. The cage was kept in darkness, and the trial ran for 24 h. Each treatment was repeated four times (N = 240) and after 24 h the number of flies caught in sticky traps was recorded to assess flight performance.

### 2.6. Field Flight Assay and Recapture Rates

This experiment assessed the recapture rates and the persistence of QD markings versus fluorescent powders in Medflies when released into an orchard. The Medfly pupae were separated into four groups consisting of 1000 Medfly pupae each, with equal sample sizes maintained across all replicates. Three groups were marked with different QD colors (yellow, red, and orange), while the control group was marked with fluorescent pink powder. The marked pupae were then incubated in 5 L ventilated containers at 25 °C with a 12L:12D cycle and provided with a sugar-water source for the emerged adults. After emergence, all treatments per replicate were placed in the center of a 1.21-hectare Nadorcott/Seedless citrus orchard at Welgevallen experimental farm in Stellenbosch (−33.9480° S, 18.8721° E) at the same time. Two delta traps were placed 30 m away on either side of the release point and about 1.5 to 1.8 m above the ground since trap height can influence recapture rates [52]. The traps were baited with Trimedlure and terpinyl acetate (ChemPack Fruit Fly Lure, Batch no. PTML 3016, Southern Paarl, South Africa), and were monitored for 72 h after each release. The experiment was repeated three times (N = 12,000), with the first two replicates taking place in Autumn and the last replicate in Spring. The weather conditions during the experiment varied across the three replicates, with average temperatures ranging from 15.8 °C to 19.2 °C. The traps were monitored and the data were collected at 24, 48, and 72 h intervals. The numbers of flyers, non-flyers, and pupae that did not emerge were also noted.

### 2.7. Transferability and Stability

This experiment assessed the transferability of the different markers on Medflies. Medflies used during this experiment were marked during the “suitable marking technique” experiment and were carried over to this experiment. Additional male and female flies were reared on blue-dyed sugar water as the control (N = 299). Marked Medflies from each treatment group (QDs on pupae, QDs on adults, and fluorescent powder on pupae) were introduced to opposite-sex flies fed blue sugar water (control group). Five marked flies from each group were placed in containers with unmarked opposite-sex flies and allowed to interact and mate for five days. The flies were euthanized and examined under a stereomicroscope with UV light to evaluate marker loss and transferability. 

### 2.8. Storage Abilities

This experiment assessed the retention and persistence of QD markings on Medflies under various storage conditions compared to traditional fluorescent powders. To determine the storage effects on QD-marked flies, Medfly pupae were placed in Petri dishes and distributed into four 2 L containers with mesh-covered lids for ventilation. The containers were incubated at 25 °C with a 12L:12D cycle until the flies emerged. After emergence, the flies were reared for two additional days under the same conditions, with granulated sugar and water provided. Once fully developed, the flies were euthanized by freezing at −20 °C, counted, and separated into four groups. Each group was then marked with either fluorescent powder (pink, yellow, or blue) or red QDs. Following marking, the flies were subjected to one of six storage treatments: storage in absolute ethanol at −20 °C, in ethanol at −80 °C, or in ethanol at room temperature (Treatments 1, 2, and 3, respectively); dry storage at room temperature (Treatment 4); or rinsing the marked cadavers in water or ethanol followed by dry storage (Treatments 5 and 6, respectively). All Medfly individuals were then examined at 2, 4, and 8 weeks under UV light to detect the presence of QDs and fluorescent powders. For each time interval, six treatments were tested with 10 Medflies each, repeated three times, totaling 180 Medflies per time interval per color (N = 2160).

### 2.9. Statistical Analysis

A range of statistical tests were used across experiments to assess the effects of marking methods on the Medfly performance parameters mentioned above. The “*suitable* marking technique” experiment used a generalized linear model (GLM) with a quasibinomial distribution to compare marking retention among the three groups. The data were transformed to proportions (values between 0 and 1) to represent retention rates, which ensured the data were appropriate for analysis under the quasibinomial distribution. This distribution was chosen because it accounts for overdispersion, particularly relevant here due to the high frequency of values at the upper limit. In the “flight ability” experiment, a contingency table was used to analyze differences in flight ability among the three treatment groups. This approach was selected because it is well suited for categorical data, allowing us to examine the frequencies of observed outcomes (e.g., flies categorized as flyers or non-flyers) across the different treatment groups. An unadjusted Fisher exact post hoc test was subsequently applied to show the significant differences among the groups. We recognize that this approach is more lenient and increases the likelihood of Type I errors. However, given the exploratory nature of this study and the preliminary evaluation of QDs as a marking technique, our priority was to detect potential differences rather than risk overlooking meaningful effects. The “recapture rate” data were analyzed using contingency tables. The “recapture rate” data were analyzed using contingency tables, for similar reasons mentioned above, enabling the evaluation of how the different marking methods influenced recapture rates over time. In the storage ability experiment, a GLM with a quasibinomial distribution was used to evaluate the retention and persistence of markings under various storage conditions across different time intervals. As with the “suitable marking technique” experiment, the data were transformed to proportions to reflect retention, and the quasibinomial distribution was applied to account for overdispersion due to frequent 100% retention rates. A post hoc analysis was performed using a Least Significant Difference (LSD) test to identify significant differences in retention rates between treatments. Finally, the transferability and stability experiment used a GLM to investigate differences in marking transferability during Medfly interactions. Again, data transformation and the application of the quasibinomial distribution ensured appropriate handling of proportional data and overdispersion. All statistical analyses were conducted using Statistica, with a significance threshold set at *p* < 0.01 across all tests.

## 3. Results

### 3.1. Suitable Marking Technique

This experiment assessed the efficacy of QDs as marking techniques for Medflies at both pupal and adult stages compared to pupae marked with fluorescent powder (Figure 1). Fluorescent powder applied to pupae was the most reliable marking technique, achieving 100% retention across all trials. In contrast, QDs applied to pupae showed the lowest retention, and QDs applied to adults had intermediate retention between the two other treatments. The GLM confirmed the significant differences in marking retention across the three treatments (Likelihood Ratio Chi-Square (LR χ^2^) = 16.45, Degrees of Freedom (df) = 2, *p* < 0.01), with fluorescent powder outperforming both QD treatments. Quantum dots applied to adult flies had higher retention rates than when applied to pupae, but both were inferior to the fluorescent powder treatment (Figure 1A). Additionally, sex did not significantly affect marker adhesion, as there was no difference in retention between male and female flies (Figure 1B) (LR χ^2^ = 0.91, df = 1, *p* = 0.34). This suggests that the adhesion and retention of both marking types were equally effective across sexes.

### 3.2. Laboratory Flight Ability

The flight performance of Medflies was evaluated across three marking techniques: fluorescent powder, QDs, and an unmarked control or reference group. The contingency table revealed statistically significant differences in flight performance among the three treatment groups (χ^2^ = 32.02, df = 2, *p* < 0.01). Subsequently, a Fisher exact unadjusted post hoc test was conducted, indicating that Medflies marked with QDs showed significantly reduced flight ability compared to those marked with fluorescent powder and the control group (*p* < 0.01). The effect size (Cramer’s V 0.37) indicated a medium effect of marking techniques on flight performance, highlighting the meaningful practical differences between the marking techniques, particularly the detrimental impact of QDs on flight performance. In contrast, no significant difference was observed in flight performance between the control group and the fluorescent powder group (*p* = 0.28), as seen in Figure 2. This suggests that fluorescent powder markings had minimal impact on Medfly flight ability.

### 3.3. Field Flight Assay and Recapture Rates

This experiment revealed notable differences in the recapture rates of Medflies marked with QDs versus those marked with fluorescent powder. The results from the contingency table indicated that QDs were ineffective as a marking method under the conditions tested, as none of the recaptured flies had QD markings. This was observed consistently across all three monitoring time points (24, 48, and 72 h), with a statistically significant difference from the fluorescent powder-marked flies when combining all three time periods (χ^2^ = 467.83, df = 9, *p* < 0.01). In contrast, a small proportion of recaptured flies, ranging from 8.5% to 9.2%, retained fluorescent powder markings. This suggests that fluorescent powder remained detectable post-release, while QDs either failed to adhere adequately to the flies or marked flies were not recaptured. Figure 3 shows an interesting trend in the proportion of non-flyers and active flyers across the marking techniques. The fluorescent powder group had significantly fewer flyers and more non-flyers compared to the three QD color groups (*p* < 0.01, Cramer’s V = 0.25). This suggests that, while fluorescent powder was effective in marking, it may have impaired flight performance, potentially reducing the mobility of marked flies and thereby affecting their recapture rates. The effect size indicates a small-to-medium effect, which means that the relationship between the marking technique and flight performance is noticeable but not strong.

### 3.4. Transferability and Stability

This experiment investigated the transferability and stability of the different markings on Medflies. Both QDs and fluorescent powder demonstrated measurable levels of marker transfer during fly interactions. Medflies marked with fluorescent powder exhibited a transfer rate of 17.8%, while those marked with QDs during the adult stage showed a similar transfer rate of 17.5%. In contrast, flies marked with QDs during the pupal stage exhibited a lower transfer rate of 10%. However, a GLM analysis revealed no statistically significant differences in transferability between the marking methods (LR χ^2^ = 2.18, df = 2, *p* = 0.34, N = 299). In terms of marking stability, however, significant differences emerged. Fluorescent powder demonstrated high stability, with no marker loss throughout the experiment, allowing for consistent identification of marked individuals. In contrast, QDs, particularly when applied at the pupal stage, showed a higher rate of marker loss over time. When looking at the results of the QD stability, the GLM showed that when QDs are applied to adults, they had significantly lower loss rates (LR χ^2^ = 8.64, df = 2, *p* < 0.01) compared to QDs applied to pupae. Specifically, Medflies marked with QDs at the adult stage (AQ) showed a mean loss rate of 51.11% (SE = 5.07, 95% CI: 41.27–60.87), whereas QDs applied at the pupal stage (PQ) exhibited a significantly higher mean loss rate of 77.5% (SE = 6.56, 95% CI: 62.25–87.8). This quantifies the disparity in marker retention and emphasizes the developmental stage’s impact on QD effectiveness. Overall, while fluorescent powder provided consistently reliable and stable marking, the efficacy of QDs depended on the developmental stage at which they were applied, with decreased stability noted when marking was performed at the pupal stage.

### 3.5. Storage Ability

In this experiment, all Medflies marked with fluorescent powder had a 100% retention rate throughout all treatments and colors, indicating the effectiveness of this marking method. In contrast, QDs did not retain their markings consistently, showing a notable discrepancy in retention compared to the fluorescent powders. These losses were observed across multiple time intervals and replicates, suggesting that QDs were not as reliably retained as fluorescent powder. A GLM was performed to determine if there were differences in marking retention rates across the treatments and over time. The analysis revealed a significant difference among the treatments (LR χ^2^ = 15.48, df = 5, *p* = 0.01), indicating that the type of treatment affected the retention of QD markings. However, there was no significant interaction between treatment and time (LR χ^2^ = 8.78, df = 10, *p* = 0.55), nor was there a significant effect of time alone on marking retention (LR χ^2^ = 2.52, df = 2, *p* = 0.28). This suggests that while the type of storage and handling treatment impacted the retention of QD markings, retention did not significantly decline over the different time points tested, although there was a general trend in the data suggesting a gradual decrease in QD marking visibility over time. The LSD post hoc test revealed significant differences among the treatments. Specifically, direct storage in ethanol at room temperature (*p* < 0.01) (Treatment 3) and rinsing the marked cadaver in ethanol followed by dry storage (*p* < 0.01) (Treatment 6) resulted in significantly lower QD retention compared to other conditions (Figure 4). 

## 4. Discussion

### 4.1. Suitable Marking Technique

The efficacy of QDs as a marking technique for Medflies appears dependent on the life stage at which they are applied. Fluorescent powder applied to pupae was found to be the most reliable marking technique, achieving a 100% retention rate in all trials, indicating its superior efficacy and consistency, which aligns with previous findings [32]. By contrast, QDs applied to pupae showed significantly lower retention rates, with statistical analysis confirming notable differences in marking efficacy across the methods employed here. This difference suggests limitations in QDs when applied at the pupal stage, possibly due to factors affecting particle adhesion, perhaps during eclosion. It is also noteworthy that, upon examining the empty pupal cases under UV light, QDs were observed adhering to the pupal casings rather than transferring fully to the emerging adults. This residual adherence onto the pupal casing may have reduced the quantity of QDs passed on to the adult flies, potentially affecting the efficacy of the marking technique. When applied to adult Medflies, QDs performed better, showing intermediate retention rates compared to fluorescent powder but still not achieving the same level of efficacy as traditional fluorescent powder. These findings indicate that while QDs have potential, particularly in marking adult flies, they currently fall short of the reliability offered by fluorescent powder, particularly if applied at the pupal stage. 

### 4.2. Laboratory Flight Ability

Quantum dots were shown to potentially have a negative impact on Medfly flight performance under controlled conditions. This reduction in flight ability suggests that QDs may compromise Medflies’ physical functions, potentially due to the size or weight of each particle [53,54,55] or their method or location of adhesion to the insect’s body. Alternatively, the application methodology used for QDs could have negatively impacted the Medflies’ flight ability instead. The application method uses ethanol, which previous studies have demonstrated impairs insect locomotion. When the QDs were applied to the adult Medflies using a 99% ethanol solution, their flight ability was affected. This observation aligns with findings that 10–20% ethanol solutions can reduce locomotion in bees [56]. Other studies that used another application method for QDs showed no effect on flight ability [42], suggesting that the combination of ethanol and the QDs could have affected the flight ability rather than the QD itself. In contrast, fluorescent powders demonstrated minimal impact on flight ability in our trials, which aligns with similar studies on insects [57] reinforcing their suitability for use in SIT programs and other pest management strategies where preserving the natural behavior of marked insects is essential.

### 4.3. Field Flight Assays and Recapture Rates

The recapture rates in this study highlighted the limitations of using QDs as a marking technique for Medflies under field conditions. While fluorescent powders showed detectable recapture rates, no flies marked with QDs were recaptured, suggesting that QDs either failed to adhere effectively or were lost during flight or after release. This result is consistent with previous findings by Gurdasani et al. (2021), who also reported challenges with QD retention in similar mark–release–recapture studies on other insect species agent [41]. The absence of QD-marked flies in recapture experiments highlights potential challenges in the practical application of this technology. While it is essential to confirm that QDs were effectively attached to Medflies prior to release, environmental factors may have also significantly influenced QD retention. For instance, exposure to UV light and high temperatures have been shown to accelerate the oxidative dissolution of QDs, releasing ions and degrading the fluorescent properties critical for detection [58]. With that being said, we do not believe photobleaching played a role in the lack of QD detection, as they have been shown to have a strong resistance to photobleaching [59] and we recaptured Medflies with fluorescent powder markings, which have been shown to have much higher rates of photobleaching than QDs [59]. However, there was an interesting trend that emerged when comparing flight activity across the marking techniques. The fluorescent powder group showed a higher proportion of non-flyers compared to the QD-marked groups, which had a greater proportion of active flyers. This observation suggests that although fluorescent powder is effective for marking, it may impair flight ability to some extent during field releases as shown by our results, potentially impacting the mobility and recapture rates of marked flies. The higher number of flyers marked with QDs in the recapture trials compared to the flight ability trials could suggest a contradiction in the results shown in Figure 2; however, this may be explained by the lack of adherence of the QDs to the Medflies, which was observed during the “suitable marking technique” experiment. This reduced adherence would mean that fewer QD-marked flies experienced flight impairment, making them more able to fly effectively and, therefore, fly a further distance away and not be recaptured in field conditions. Another possibility as to why there was a higher number of flyers marked with QDs is that they were initially able to take off, but may have struggled to sustain flight over longer distances due to the QDs. This could have prevented them returning to the traps due to flight impairment caused by the QD application method, which is seen in Figure 2. Additionally, the higher proportion of non-flyers in the fluorescent powder group could also be explained by the increased self-grooming behavior when marked, as tephritid flies are known for their persistent grooming behavior, which leads to most superficial dye particles being removed from their bodies [18,30]. Fluorescent powder markings are often more visible and extensive, as observed during our marking process. The substantial coverage could have stimulated more frequent grooming activity, which could have increased the likelihood of the Medflies staying stationary, contributing to the higher proportion of non-flyers in this treatment. However, further research will need to be performed to confirm this relationship, as this is just a theory.

### 4.4. Transferability and Stability

Both marking techniques showed some level of marker transferability during interactions such as courtship, but no significant differences were observed in the transferability rates when comparing QDs to fluorescent powder. However, fluorescent powder demonstrated better stability, retaining its original markings fully throughout the experiment. This reliability ensures consistent identification of marked individuals, which is essential for accurate population tracking in field studies. In contrast, QDs showed significantly higher rates of marker loss over time, particularly when applied at the pupal stage. This finding aligns well with previous research [42], identifying challenges with QD retention, and emphasizing the limitations of QDs for long-term mark–release–recapture experiments. The instability of QDs at the pupal stage is especially problematic, as Medflies are often marked at this stage for practicality in mass-rearing and transportation prior to release. This instability of marker retention likely jeopardizes the accuracy of long-term monitoring in SIT programs, where reliable marking over time is essential. This may have been the reason why no QD-marked medflies were recovered during the recapture rates experiment. Further, a comparison between QDs applied to adults versus pupae indicated that adult-stage marking yielded greater stability, suggesting that the developmental stage at which QDs are applied influences their retention. While fluorescent powder provided robust, stable marking regardless of life stage, the effectiveness of QDs with our application method was life-stage-dependent, with reduced stability in pupal applications. These findings highlight the need for refining the QD application method to enhance their reliability if it is to be effectively used in the SIT and related field studies.

### 4.5. Storage Ability

The storage ability experiment showed significant differences in the retention and durability of markings between fluorescent powders and QDs under various long-term storage conditions. Fluorescent powders exhibited a 100% retention rate across all treatments, with no significant marker loss even after prolonged storage under diverse conditions, including freezing and dry storage. This durability is crucial for long-term studies and mark–release–recapture programs, where consistent marking visibility is essential for tracking insect populations. These findings align with prior research by Clymans et al. (2020), which demonstrated that fluorescent powders maintain stability across various storage conditions, including alcohol, wet, and dry environments [32]. This stability makes fluorescent powders a preferred choice for extended insect studies. In contrast, Medflies marked with QDs showed variable retention rates, with significant differences in marking retention found across the treatments. Although time alone did not significantly affect QD retention, storage of the marked Medlfies in ethanol at room temperature, and rinsing the cadavers in ethanol before dry storage, resulted in significantly lower retention rates. This indicates that the type of storage treatment and handling procedure is critical to QD stability, though a general trend suggested a gradual decrease in visibility over time. These findings emphasize the need for further refinement in the application method and stability of QDs, especially if they are to be considered a viable alternative to fluorescent powders in long-term mark–release–recapture studies. The lower retention and sensitivity of QDs to certain storage conditions suggest limitations in their current formulation, reinforcing the reliability of fluorescent powders for applications where durable, visible marking is essential.

### 4.6. General QD Discussion

There have been concerns with QDs with regard to the environmental and toxicity issues, including those used in biological and agricultural applications. Quantum dots can release metal ions which generate reactive oxygen species (ROS), potentially leading to oxidative stress and DNA damage [60]. While the Copper Indium Disulfide/Zinc Sulfide (CuInS_2_/ZnS) QDs used in our study are considered less toxic and more environmentally sustainable than cadmium- or lead-based QDs, which release harmful Cd^2+^ or Pb^2+^ ions [61], their overall safety is unknown. For example, a study investigating the toxicity of hydrophilic green cadmium telluride/cadmium sulfide (CdTe/CdS) QDs against *Tribolium castaneum* found no significant side effects on beetle behavior or survival [42]. Although CuInS_2_/ZnS QDs are marketed as safer alternatives, their potential for bioaccumulation and long-term environmental impacts, particularly when used at scale, still needs to be investigated. These risks, including potential harm to non-target species or ecosystems, highlight the need for thorough safety assessments and mitigation strategies to ensure their responsible application in pest management and other fields.

The impact of ethanol and QDs on insect flight ability highlights the importance of exploring alternative ethanol-free application methods. This has been performed before, with studies on *Tribolium castaneum* showing that QDs can be effectively applied without ethanol by instead immersing insects in hydrophilic CdTe/CdS QD solutions stabilized with mercaptosuccinic acid (MSA) and dispersing them in water. This method involves short immersion times followed by air-drying, allowing the QDs to adhere without the need for solvents that could negatively affect behavior [42]. Similar techniques have been successfully applied to Queensland fruit flies and diamondback moths, suggesting that ethanol-free approaches could be adapted for Mediterranean fruit flies. Implementing such alternative methods may help mitigate the negative effects of ethanol; however, this methodology may not be feasible for the large-scale marking and release of Medflies in SIT programs. Future studies should focus on evaluating the feasibility and scalability of these other techniques for mass-marking applications.

The lack of adherence of QDs to Medflies noted in the study could be attributed to the specific marking technique and solvent used in this study. The application method relied on ethanol as a solvent, which, due to its rapid evaporation, might not have allowed sufficient time for the QDs to bond effectively to the insect cuticle. This marking technique contrasts with other studies where QDs demonstrated successful adherence to insect surfaces, suggesting that alternative solvents or application methods may better facilitate QD retention [42]. Additionally, the physical characteristics of the marking process, such as the method of agitation and the life stage at which the QDs were applied, likely influenced adhesion. For instance, applying QDs to pupae might have reduced their effectiveness as some of the QDs remained on the pupal casing during eclosion, limiting transfer to the adult flies. These factors highlight the need for further investigation into optimizing marking techniques and solvent formulations to improve QD retention if it were to be used in marking Medflies. Future work could consider (i) altering the functional properties of QDs to improve attachment to the insect’s cuticle, and perhaps especially tailored to Tephritidae; (ii) alternative chemicals to dissolve the QDs and apply them to insects with lower potential impacts on the flies’ biology than ethanol so it can be used in mass-marking; and (iii) further consideration of other techniques for marking fruit flies for enhanced pest management [50].

## 5. Conclusions

In conclusion, while QDs offer an intriguing potential alternative to traditional fluorescent powders for marking Medflies, this study shows that QDs currently fall short in several critical areas. Fluorescent powders consistently outperformed QDs in terms of marking retention, their impact on flight ability, recapture rates, and long-term stability, making them the better option for SIT programs and other pest management strategies that rely on effective insect marking. Although QDs have potential, especially given their versatility and small size, their use in entomological studies requires significant refinement to address the issues identified in this study. Future research should focus on optimizing QD application methods and improving their retention and stability over time to be able to use their full potential as a marking technique for Medflies.

## Figures and Tables

**Figure 1 insects-16-00270-f001:**
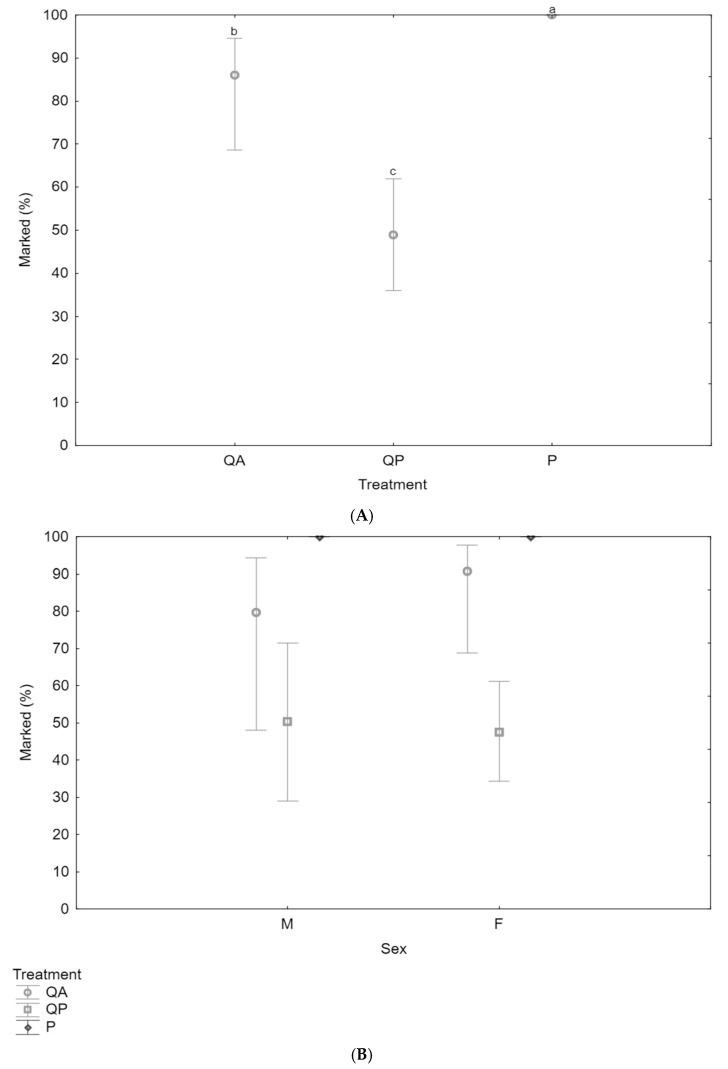
(**A**) The percentage of Medflies marked after 8 days using three marking techniques: QDs applied during the adult stage (QA), QDs applied during the pupal stage (QP), and fluorescent powder applied during the pupal stage (P). A total of 240 Medflies (N = 240) were used across the experiment, with each treatment group consisting of 20 specimens per replicate, repeated four times. Vertical bars indicate 95% confidence intervals. Significant differences between treatments are denoted by different letters (a, b, c), where *p* < 0.01, indicating that marking retention varied significantly between the groups. (**B**) Comparison of the marking percentages across sexes (male and female) for the different marking treatments. The same three marking techniques were evaluated, with males and females analyzed separately to assess potential sex-based differences in marking retention. Vertical bars represent 95% confidence intervals.

**Figure 2 insects-16-00270-f002:**
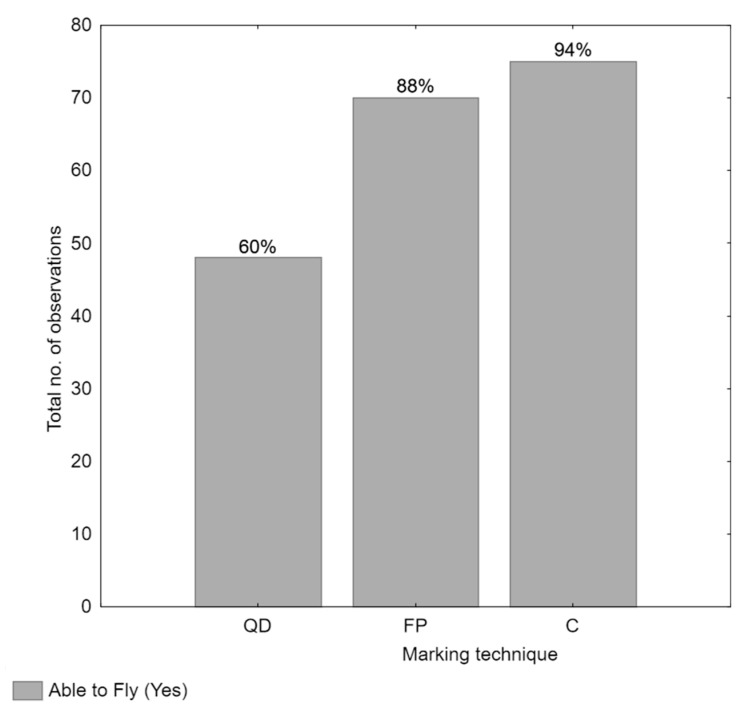
The total number of Medflies able to fly under different marking treatments when attracted by a food and light source. Medflies were marked using one of two techniques: QDs applied during the adult stage (QD) or fluorescent powder applied during the pupal stage (FP). A control group (C) of unmarked Medflies was also included for comparison. A total of 240 Medflies (N = 240) were tested in a controlled flight trial, with each treatment group consisting of 20 specimens per replicate, repeated four times. Vertical bars indicate the number of observations, with the percentage of individuals from each marking treatment that could fly displayed on each bar.

**Figure 3 insects-16-00270-f003:**
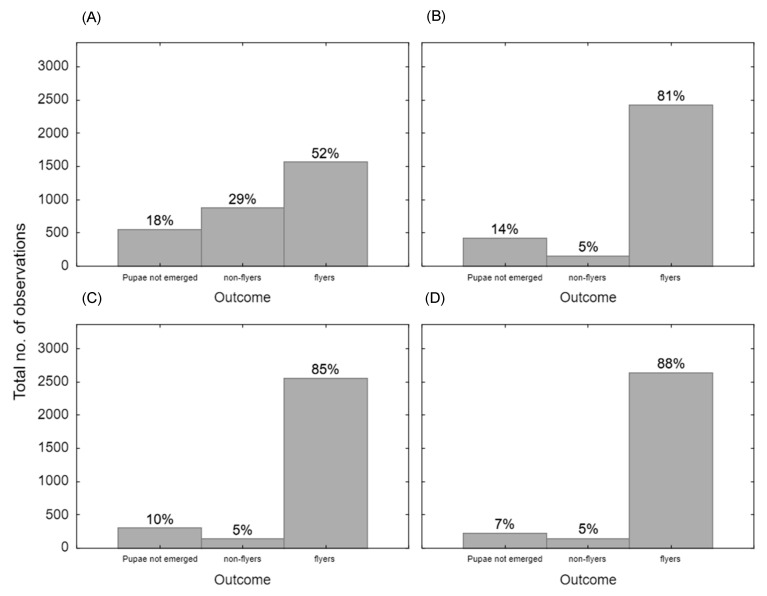
The total number of Medflies combined at 24, 48, and 72 h, categorized as flyers, non-flyers, or those that did not emerge during the mark–release–recapture study. Medflies were marked with either fluorescent powder (**A**), yellow QDs (**B**), red QDs (**C**), or orange QDs (**D**) during the pupal stage to assess the persistence and effectiveness of QD markings in comparison to fluorescent powder. A total of 12,000 Medflies were marked and released (N = 12,000), with equal distribution across all marking groups (3000 Medflies per group, repeated across three experimental replicates). Bars indicate the total number of observations, with percentage breakdown of the different categories displayed on each bar. Environmental conditions during the study varied across replicates, with temperatures ranging from 15.8 °C to 19.2 °C. The experiment was conducted over two seasons (Autumn and Spring), allowing for assessment under different climatic conditions.

**Figure 4 insects-16-00270-f004:**
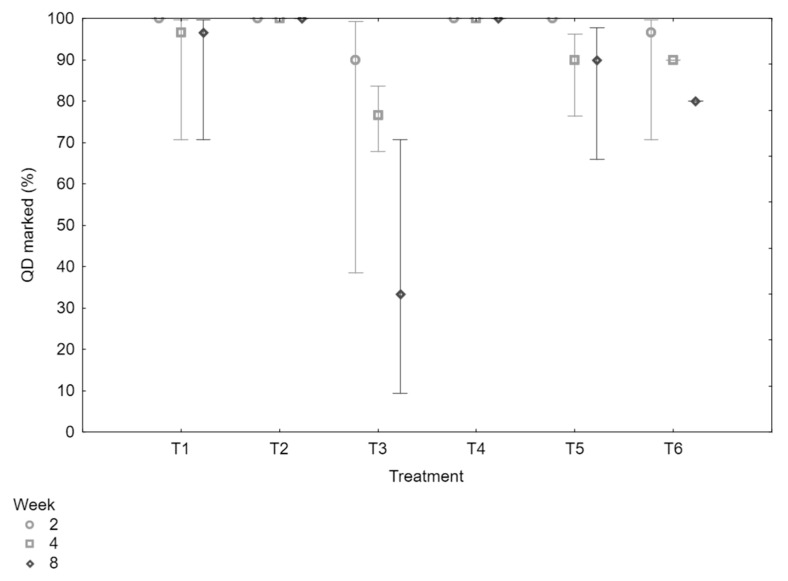
The percentage of Medflies that remained successfully marked with QDs during the six storage treatments (T1–T6) across all time intervals (2, 4, and 8 weeks). The treatments represent different storage and handling conditions, including storage in ethanol at different temperatures (−20 °C (T1), −80 °C (T2), and room temperature(T3)), dry storage at room temperature (T4), and rinsing in water (T5) or ethanol (T6) before dry storage. This experiment aimed to assess the retention and persistence of QD markings on Medflies under different storage conditions, compared to traditional fluorescent powders. The fluorescent powder retained 100% of its markings across all treatments and time intervals, and was therefore not included in the graph. For the entire experimental study, a total of 2160 Medflies were used, evenly distributed across all six treatments and time periods (N = 2160). Vertical bars indicate 95% confidence intervals. Retention of markings was evaluated under UV light, with flies examined at each time point to determine the stability of QDs and fluorescent powders over time.

## Data Availability

The original contributions presented in this study are included in the article; further inquiries can be directed to the corresponding author.

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
