# Peer review of "Preliminary Analysis of Quantum Dots as a Marking Technique for Ceratitis capitata"

_insects, 2025, doi:10.3390/insects16030270_

Round 1

Reviewer 1 Report

Comments and Suggestions for Authors

please see my major and minor comments in the attached PDF. 

Author Response

Thank you for your thorough and insightful review of our manuscript. We appreciate the time and effort the referees have taken to provide detailed and constructive feedback. In the following response, we address each of your comments systematically.. Where possible, we have provided additional discussion points to address the major and minor concerns. We believe the revisions have strengthened the manuscript and improved its contribution to the field. Below, we detail the specific changes made in response to each of the comments raised.

We hope you find this acceptable and look forward to hearing from you.

Sincerely,

Dillon, on behalf of all authors

1a. The premise of the study is not clear. I understand that this is a preliminary analysis but this does not preclude from providing this appropriate rational and knowledge gap. Specifically, the introduction needs to more clearly articulate the specific advantages QD offer over existing methods beyond their small size and versatility, both being a common feature between QD and FP at least. What specific limitations of current techniques do QDs aim to address? 

>>>RESPONSE: We thank the reviewer for this constructive comment. To address this, we have expanded and reworded the introduction to better articulate the premise of the study and address the knowledge gap. Specifically, Lines 127–129 in the revised manuscript now highlight the need for an alternative, easier method to mark small flying insects, based on the negative side effects of fluorescent powders listed above. Additionally, we have reworded and expanded Lines 109–120 of the original manuscript (now Lines 133–144 in the revised manuscript) to provide a clearer rationale for exploring QDs as an alternative marking method. This includes detailing their unique optical properties, photostability, and the ability to tune their emission spectra, which address key limitations of fluorescent powders. We hope this revised introduction provides the necessary clarity and rationale as requested.

1b. Line 36, the authors state "QDs possess potential as marking agents, particularly given their versatility, specificity and colour range [...]". Please expand the introduction to answer the following questions: 

  1. What does versatility mean here? 
  2. Specificity to what? Different types of insects? 
  3. Why is the colour range available for QD important? Is there any fly tracking experiment that would benefit from the wide range of color

>>>RESPONSE: The sentence originally starting at Line 36 in the original manuscript ("QDs possess potential as marking agents, particularly given their versatility, specificity, and colour range [...]") has been removed in the revised version. The mention of versatility, which was in reference to the various applications of QDs that are already being used, has also been removed. However, we have expanded and clarified some of the removed abstract points from the original manuscript in the revised introduction. This includes the addition of Lines 151–153 in the revised manuscript and the rewording of Line 122-123 in the original manuscript (now Lines 155–156 in the revised manuscript) for improved clarity on what we were referring to as specificity. The discussion on colour range and its advantages has also been expanded in the revised manuscript in Lines 136–144, which should help clarify why the color range and other advantages are important for QDs.

  1. The authors must do more control experiments. The control used in Fig 2 does not seem appropriate because the QD staining procedure uses 99% ethanol (which seems unnecessarily aggressive to apply to a fragile specimen like a small fly). Ethanol will almost certainly impact the fly development from pupal to adult stages. In the discussion, the authors actually acknowledge the detrimental effect of ethanol, citing a study in bees (line 463). How did the adult fly behave after being exposed to 99% ethanol? Please add a control for QDs which consist of doing exactly the same procedure as QD (same incubation time, rinsing steps, solvent), but with 99% ethanol only (without the QD). What was the solvent for the FP? Same as QD? If not, the authors need to do another control for FP using FP-specific solvent. Please describe the control steps in the Methods and edit all necessary figures and statistics. 

>>>RESPONSE: Thanks for raising this issue. This is a fair point and the referee is indeed correct that we do not know precisely if it is the QD or the ethanol + the QD that affected the outcomes of our trials. However, this was not the purpose of our experiments and it is not essential to know the answer to that question to address the objectives of the current paper. While it would perhaps be interesting to know if it’s the ethanol solution causing any negative effects on fly performance, there simply aren’t many alternative ways to dose the flies with QD at present as the QD must be dissolved in a suspension of either ethanol or hexane for our objectives. Ethanol was selected as the less harmful of the two options as hexane is known to severely impact cuticular hydrocarbons and lipids. Furthermore, the application we used would likely be the ‘standard practise’ if QD were to be applied en masse in a SIT program and so we’re comparing two alternative ‘standard’ approaches in a manner that is relatively straightforward to interpret for practitioner’s in the field. We think this is useful and transparent. We acknowledge now in the revised ms Discussion that it is possible the ethanol + QD solution can be hampering fly performance rather than the QD themselves, but alternative application methods would need to be developed and was beyond the scope of the present paper (Lines 1117-1119  in the revised manuscript). We hope these changes are satisfactory.

  1. Data related to Fig 2 and 3 are contradictory. Fig 2 argues that FP is better than QDots in terms of flying ability. Data in Figure 3 argue the opposite. Therefore, please revise this sentence (1. 349), as this is likely not true based on your data in Fig 2 "while fluorescent powder was effective in marking, it may have impaired flight performance". The discussion 4.2 will need some attention after the authors fix the contradicting result on fly impairment between Fig 2 and Fig 3 

>>>RESPONSE: Thanks for raising this point, however we politely disagree with the referee’s interpretation that there is a gross contradiction. These are two very different assays that are performed in very different operational environments each with their own sets of distinct confounding factors to interpreting ‘performance’. Note that in the original ms we had discussed this apparent discrepancy in the outcomes of the two different assessment techniques for flight ability (Lines 487-492 of original ms, now Lines 1205-1228 on revised ms). Typically, lab flight assays assess willingness to disperse as well as flight ability (akin to physiological performance) in a closed system, while field assays are an open system and thus inherently have much more variability, including a high disappearance (loss) rate of released flies whether marked by pigment or another marking technique (see e.g., discussion in Steyn VM, Mitchell KA, Nyamukondiwa C, Terblanche JS. Understanding costs and benefits of thermal plasticity for pest management: insights from the integration of laboratory, semi-field and field assessments of Ceratitis capitata (Diptera: Tephritidae). Bulletin of Entomological Research. 2022;112(4):458-468. doi:10.1017/S0007485321000389). No-one really knows what happens to the missing flies. It could be they lose their markings, that they die from various causes during dispersal, or that they are eaten by a predator, among several options. However, drawing on our other trial results presented in the ms we suggest that QD do not adhere well, and so it is reasonable to expect low QD-marked fly recapture rates in field assays that occur over longer timescales than the lab flight assays, while a shorter assay will have higher QD adherence. Furthermore, lab flight assays capture other aspects of behaviour such as innate drive to move, or being shorter duration assays perhaps the flies spend some time undertaking other behaviours (e.g., grooming, interacting) in the lab which temporarily suppresses realized movement assayed over a shorter period. We think this is both a plausible and reasonable interpretation based on our observations and the data at hand. We take care to explain these points in the revised ms carefully (Line 732-736 in revised ms).

4a. Statistical analysis - The study employs a variety of statistical techniques, such as Generalized Linear Models (GLM), non-parametric Kruskal-Wallis tests, and post-hoc analyses for comparing markers retention or flight performance metrics. However, the manuscript lacks detailed information on sample sizes for some tests (e.g., field trials), which could affect the reliability of results. The non-parametric tests could be complemented with effect size metrics for better interpretation of practical significance.

>>>RESPONSE: Thank you for this point. To address these concerns we made the following changes: A correction that has been made to the manuscript is that the ‘flight ability test’ was analysed using a contingency table and has been updated in the revised manuscript (Line 626 in revised ms; Line 250-251 original ms). The sample sizes and a more detailed explanation of the specimens used have been included in the revised manuscript as follows: for the Suitable Marking Technique experiment (line 433–434 in revised ms), the Laboratory Flight Assay (lines 438 and 446 in revised ms), the Field Flight Assay and Recapture Rates (lines 451-452 and 594 in revised ms), the Transferability and Stability experiment (line 505in revised ms), and the Storage Abilities experiment (lines 527-529 in revised ms). Both non-parametric tests are now complemented with effect size metrics to help interpret the practical significance. This can be seen (line 748 and line 822) in the revised ms.

4b. It is unclear if data normalization was applied to meet assumptions for GLM or other parametric tests.

>>>RESPONSE: Apologies, we should have been clearer on these issues. The point regarding data normalization for the GLM and other parametric tests is valid. To clarify, the data was transformed to values between 0 and 1 to represent proportions. This transformation ensured the data were suitable for analysis under a quasibinomial distribution, which is well-suited for proportional datasets that often exhibit overdispersion. In this study, many data points ranged between 0 and 100%, with a high frequency of values at the upper limit (100%), contributing to the overdispersion. The quasibinomial distribution accounts for this variability, allowing for a more robust and accurate analysis of the data. Given this transformation and the proportional nature of the data, the quasibinomial distribution was an appropriate choice for the GLM, allowing for a robust analysis without requiring further normalization. We have updated the statistical analysis section in the revised manuscript to explicitly describe this approach as seen in line 537-650. We hope this clarification adequately addresses the reviewer’s concern, and we are happy to provide further details if needed.

4c. More robust methods, such as Bonferroni corrections or Tukey's HSD, should replace unadjusted Fisher tests.

>>>RESPONSE: Thanks for raising this point, however we politely disagree with the referee’s suggestion. Our reasoning is based on the balance between Type I and Type II errors that occur during hypothesis testing. While methods such as Bonferroni corrections or Tukey’s HSD provide strong protection against Type I errors, they do so at the cost of increasing the likelihood of Type II errors, particularly in studies with smaller sample sizes or exploratory analyses such as ours. Given the preliminary nature of this study, our priority was to be on the side of being more inclusive in detecting potential effects, even at the risk of some false positives. We decided to do this in order to not miss any meaningful patterns. We have now clearly stated in the manuscript that the use of unadjusted Fisher tests increases the likelihood of Type I errors and that findings must be interpreted with caution (line 632-635).

4d. Please include effect size calculations and confidence intervals to complement p-values, and provide sample sizes for all experiments.

>>>RESPONSE: Please refer to our earlier response in #4a for details regarding sample sizes. Effect size calculations and confidence intervals have been included where relevant to the experiments. If additional details are required to fully address the reviewers' concerns, we would be happy to provide further information.

4e. When studying the flight ability, KW test is used, whereas GLM is used for the optimum marking technique. The authors need to be consistent in how the statistical significance is computed or they need to describe the rationale for the method used in each specific case. The statistical analysis section describes what test was used but not why it was used in each particular experiment. Please define LR, df meaning.

>>>RESPONSE: Thank you for picking up on this. There was typographical error, as KW test was not used for ‘flight ability’ but rather a contingency table (line 627 in revised ms; in original ms). The rationale behind using the contingency tables for two experiments and GLM for the remainder is due to the type data and the objectives of each analysis. Contingency tables were used in experiments like 'flight ability' and 'recapture rates' because these analyses involved categorical data. This allowed us to evaluate differences in observed frequencies between treatment groups in a straightforward manner.  For the remaining experiments, GLMs with a quasibinomial distribution were used. This n was based on the proportional nature of the data and the need to account for the high overdispersion. The data was transformed to values between 0 and 1 to represent proportions, ensuring they were suitable for analysis under the quasibinomial distribution. Overdispersion was particularly relevant in our cases due to a high frequency of values at the upper limit.

  1. The discussion repeats findings without critical evaluation of broader implications. 

For example, while QDs are highlighted as a future option, their fundamental shortcomings are not sufficiently addressed. Here is a list of points I would like the authors to discuss

5a. The authors could better describe the potential improvement that would enable the QDots tagging approach to be more effective. Maybe chemical functionalization? Maybe using a carboxylic group activated surface for better attachment? 

>>>RESPONSE:  Please also see our earlier response in #1b. Altering the functional properties of QDs, such as through chemical functionalization or the addition of a carboxylic group, is indeed a valid and promising approach to improve attachment to the insect’s cuticle. However, this area falls outside the scope of the authors’ current research expertise, as well as the scope of the current paper, so we are unable to provide a detailed exploration of these techniques. To acknowledge this, the idea of modifying QD properties has now been briefly mentioned in the revised manuscript on lines 1375-1376.

5b.  The potential toxicity of QDs to Medflies and the environmental impact of their use should be discussed. QD nanoparticles are notoriously toxic, not only for the specimen but also likely for the environment. The authors should discuss the viability of a QD approach used at scale, and the potential impact on the environment and local animal species feeding on flies or other tagged insects. 

>>>RESPONSE: We agree that this is a topic that should be discussed and has now been added to the revised manuscript line 1314-1327

5c. The authors found that QDs significantly impair flight ability, a critical drawback for their use in pest management. The authors connect this impairment to ethanol use during application, which aligns with existing literature on ethanol's effects on insect locomotion. However, the authors could have explored or at least discussed alternative application methods for QDs to mitigate ethanol's negative effects

>>>RESPONSE: We agree that this is a topic that should be discussed and has now been added to the revised manuscript line 1328-1340

5d. The absence of QD-marked flies in recapture experiments is a critical failure for practical applications. However, it is unclear to me whether the author checked that QD was effectively attached to the Medfly prior to release. Moreover, could the author discuss the extent to which environmental factors (e.g. weather, predation, humidity, temperature) affect QD retention?

Thank you for bringing this up. The attachment of the QDs to the pupae was briefly assessed prior to the release; however, inspecting all 4,000 flies individually per replicate would not have been feasible. While a subsample could have been examined, this would not have significantly altered the outcome due to the inherently low recapture rates typical of fruit fly release-recapture experiments. The absence of QD-marked flies in recaptures is consistent with findings from the marking experiment, where QD retention on pupae was the lowest. This suggests that the result is indeed an artifact of the initially low QD. While weather conditions would indeed be an interesting factor to consider, we believe this falls outside the primary scope of the current study. However, we acknowledge its potential significance and agree that it warrants further investigation in future research. To address this valuable point, we have included a brief discussion in the revised manuscript (lines 1191–1199).

5e. Discuss next steps to refine the QD-marking technique, like investigating alternative solvents that do not impair insect behaviour, different nanoparticle coatings to enhance adhesion and reduce loss during flight. Discuss other tagging protocols to improve retention and visibility, and other fluorescent markers (fluorescent proteins or nanodiamonds). How about genetically modified sterile fly expressing a genetically encoded fluorescent protein, a widely used technique in biology? This can be discussed as well: the protein is non toxic, the sterile fly will not pass down its transgene so the natural ecosystem will not be impacted, the fluorescent maker will not disappear over time or during flight or mating because it expressed by the cell, intracellularly, instead of extracellularly for QD and FP. As a non-expert in this ethology field, that seems like a viable and tractable alternative. Please broaden the discussion.

>>>RESPONSE: In the revised manuscript, we have addressed these points in the discussion by highlighting potential improvements and future directions. Specifically, we note that the lack of adherence of QDs to Medflies could be attributed to the marking technique and the solvent used (line 1341-1375 in revised ms). There have been many successful other marking techniques such as fluorescent proteins or staining flies’ food artificial diet with Chalco red, however we believe that this is outside the scope of the study. Our primary focus is on evaluating QDs as a marking method. However, we recognize the value of these techniques and have included a brief section at the end of the discussion to address potential future directions (line 1375-1380 in revised ms)

  1. The term 'transference' does not seem adequate. Unless it is widely accepted in the field, I would prefer transfer or transferability, as transference is primarily used in psychology to describe the redirection of feelings from one person to another. 

>>>RESPONSE: Done. We have changed all the necessary wording to transferability instead of transference to be clearer in the intended meaning.

  1. Line 95: "Different powder colours can be hard to differentiate” better explain why the field needs this multispectral feature. Why would it be different for QD? 

>>>RESPONSE: Thank you for this point and we agree that it needed to be more clear as to why QD would be different. In the revised manuscript, we have clarified these points to highlight the advantages of QDs as marking technique, compared to fluorescent powders. We highlight the fact that QDs offer a broad excitation spectrum and narrow emission bands, which allow their fluorescence colors to be precisely tuned by modifying their composition and size (Line 138-139 in revised ms). This allows for a wide range of distinct and easily distinguishable colors, which is particularly valuable in field studies requiring multispectral differentiation or multiple treatments. For example, in mark-release-recapture experiments, the ability to use multiple distinct colors allows researchers to track different insect populations, release groups, or temporal cohorts within the same study area without overlap or confusion. The photo-stability of QDs further enhances their utility, as their fluorescence remains strong and reliable even under prolonged exposure to UV light compared to traditional fluorescent powders (line 140-142 in revised ms).

  1. Line 122: “ability to be modified to suit specific insect species." How is this done? With specific surface chemistry?

>>>RESPONSE: This information was deduced by Gurdasani et al (2021), which stated “Additionally, the CdTe/CdS QD could also be used for other insects like the Queensland fruit fly and have a broad scope for modifications to suit a wide variety of insect exoskeletons (Figure S1, supporting information).” The revised manuscript has been slightly adjusted to improve clarity (line 151-153 and line 155-156)

(Gurdasani, K.; Li, L.; Rafter, M.A.; Daglish, G. J.; Walter, G.H. Nanoparticles as potential external markers for mark–release–recapture studies on Tribolium castaneum. Entomol. Exp. Appl. 2021, 169(6), 575-581.)

  1. Clarify the specific criteria used to determine "optimal" marking. How was marking retention quantified? 

>>>RESPONSE: Thank you for raising this point, as it brought to our attention that the title of this section may not have been appropriate. In response, the chapter title and further references of “optimal marking technique” has been changed to “suitable marking technique” in the revised manuscript to better reflect the scope and findings of this experiment.

Marking retention was quantified by visually inspecting marked flies under a stereomicroscope equipped with UV light. Each marked individual was assessed for the presence or absence of visible markings, and the retention rate was calculated as the proportion of marked flies out of the total examined.

  1. The authors need to provide more essential information about the QD; the product ref alone is not sufficient. Please state QD size, core/shell composition, surface coating, and excitation/emission wavelengths 

>>>RESPONSE: Agreed. We have now updated the methodology section to now include all the necessary information (line 313-320 in revised ms; line 165 in original ms)

  1. Acronyms should be defined at first use (GLM in line 247). Moreover, did the author mean General or Generalized? 

>>>RESPONSE: Done, as suggested. We have done so now where GLM (line 249 of original manuscript) is defined as a generalized linear model (line 538 in revised ms).

  1. In methods, it would be better to compare the fluorescent marker in the same unit, either uL or ug, so that the reader can quickly compare the quantity used. The request is relevant because QD and FP are comparable in their nanometer size. 

>>>RESPONSE: Thank you for picking up on this, and we agree that they should be in the same unit in order to compare quickly. This is shown in the revised manuscript (line 322) where the mg amount of QDs applied per 1000 medfly is shown which can be compared to the amount of fluorescent powder applied to 1000 medfly (line 302 in revised ms; line 158 in original ms)

  1. While the size of QD and FP is comparable (~ few nm), QD nanoparticles tend to aggregate if the solvent is not appropriate, thereby losing their 'minute size' promising feature. Did the authors assess whether QD were aggregated prior to marking the fly (pictures in Figure 6 show clear aggregation)? What steps were taken to ensure uniform application and minimize QD aggregates?

>>>RESPONSE: Good point. To minimize QD aggregation, we spun the QD solution in a standard benchtop centrifuge to separate any aggregated particles and gently agitated the solution prior to pipetting to ensure even dispersion. Following this, the QDs were applied to pupae or adults in a Petri dish, which was shaken to further distribute the QDs evenly before the ethanol solvent was allowed to evaporate quickly (line 412 - 419 in revised ms; line 169-176 in original ms). These steps were implemented to improve uniformity in QD application and minimize the impact of aggregation. Although the method was not perfect, it was at the time the best solution we had to minimize the aggregation issue.

14.Line 472: "either failed to adhere effectively or were lost during flight or after release." Other reasons might be that QD photobleached, or QD were not attached from the get go? Did the authors checked that the Medflies were adequately labeled before release? How many were marked before release?

>>>RESPONSE: We do not believe that photobleaching played a role in the lack of QD-marked flies detected in the recapture experiments. Quantum dots are known for their strong resistance to photobleaching, with studies showing much lower rates of photobleaching compared to traditional organic dyes (Zhu et al., 2013; Bailes, 2020). In terms of the markinbg of the Medflies prior to release, please refer to answer #5d

  1. Why are some of the error bars asymmetric? (e.g. Fig 1). Please also indicate the number of observations N.

>>>RESPONSE:The error bars are asymmetric as the data is not normally distributed, hence why the data was transformed. The number of observations (N) is now indicated in the caption for figure 4 (previously figure 1)  (line 724in revised ms)

  1. Did the author assess the death rate of the QD vs FP (vs proper control vs naive fly) marking strategy? I believe an increased death rate post incubation will affect the recapture rate. The authors should provide more details on the QD application process, particularly in relation to the observation that QDs adhered to pupal casings rather than transferring to adult flies.

Thank you for bringing up, as we believed it was also an important point to investigate. We did attempt to assess the death rate of the QD vs FP but failed to collect sufficient data to analyse statistically.

We think the application technique explanation has been improved, not only in the ‘Quantum dots’ section, but for the remainder of the methodology section. Please let us know if you require more information.

17) Figure 2 should be better presented: merge all 3 plots in 1 with 3 different color or maker type or gray scale, so that the reader can very quickly grasp the difference between the 3 making techniques. Please add the ethanol-only control as well, and add error bars and statistical significance. Replicate this experiment to compute the error bar. Discuss whether the inability to recapture QD-fly is due to the flying impairment of QD (data from Figure 2A). Fix type: "Tecnique" > technique

>>>RESPONSE: Thank you for the suggestion. Figure 5 (which was previously figure 2) has been revised to display all plots within a single graph for improved clarity and comparison. The decision not to conduct additional trials has been addressed in response to Comment 2. Additionally, the discussion has been updated to acknowledge that the inability to recapture any QD-marked flies may be attributed to flight impairment caused by the QDs (line 1225-1229 in revised ms)

18) Be consistent with QDs vs quantum dots throughout the text. Once the acronym has been defined, stick to it. 

>>>RESPONSE: Done, as suggested.

19) Describe the statistical test used in 3.3 / figure3, with number of samples. 

>>>RESPONSE: The manuscript has subsequently been changed to explain that a contingency table was used to get our results (line 813-814 in revised ms; line 340 in original ms)

20) The following sentence is unclear (line 129). Please revise. "This outlined by having a straightforward application process and that the QDs that remain stable and securely attached under various environmental conditions without interfering with the organism's natural behavior." 

>>>RESPONSE: Thank you for mentioning this point and we agree, that line 129 needs revision. It has subsequently been changed to:

“These criteria are outlined by having a straightforward application process that remains stable over time, as well as the ability to remain securely attached under various environmental conditions without interfering with the organism's natural behavior. “ 

This can be seen in the revised manuscript (line 269-272)

21) To what extent does fluorescence photobleaching (due to prolonged sunlight exposure for example) affect the field flight assay or the overall technique, for each of QD and FP.To what extent would fluorescent nanodiamond, a perfectly photostable ~10 nm size NIR fluorescent nanoparticle, be an alternative solution?

>>>RESPONSE: As explained earlier, we do not believe photobleaching significantly impacted the field flight assay, as the longest time interval was only 96 hours. Quantum dots are well-documented for their strong resistance to photobleaching, with studies indicating much lower rates compared to traditional organic dyes (Zhu et al., 2013; Bailes, 2020). While some degree of photobleaching may have occurred in the fluorescent powder markings, the successful recovery of fluorescent powder-marked flies suggests that photobleaching was not a determining factor in the lack of QD-marked fly recaptures.

With regards to the fluorescent nanodiamond, although this may be a viable or potentially better alternative solution, this falls outside the scope of the study.

22) More often than not, it is unclear when the authors used GLM or KW to compute LR/df values and statistical significance. Please specify for each or use the same statistical method throughout the study (see major comments). 

>>>RESPONSE: Thank you for raising this point. We have now explicitly stated the statistical methods used in each results section and provided a comprehensive overview of the statistical approach in the methodology section under ‘statistical analysis’ (lines 533-646 in revised ms; lines 247–263 in original ms) to ensure clarity to what statistical test were used

23) Line 342: "a small proportion of recaptured flies, ranging from 8.5% to 9.2%, retained fluorescent powder markings." Instead of stating the range (min/max? Quartile? percentile?), please provide more rigorous statistics: mean/C195, plus number of samples over which the C195 was computed.

>>>RESPONSE: Thank you for raising this point, however, we don’t have sufficient replication to perform rigorous reporting of these statistics and they are not central to the paper’s conclusions.

24) Line 374-376: provide value for QD and FP, near the statistical test. Also state which test was used and the sample N. 

>>>RESPONSE:The manuscript has been updated to include the values for QD and FP, along with the type of statistical test used (lines 971-977in revised ms; lines 374–376 in original ms). Additionally, the sample size has now been specified in these lines.

25) Some plots could be removed and values added directly to the text. For example, instead of Fig 4, just add mean/C195 in line 383-384. 

>>>RESPONSE: In response, we have removed Figure 4 (from the original manuscript) and integrated the relevant data into the text (lines 984–991 in revised ms) to improve the readability, and overall look of the study.

26) Line 528 fixe typo: "ethonal” 

>>>RESPONSE: Thank you for bring this to our attention, it has since been changed (line 798-799  in revised ms; line 528 in original ms)

27) The figure should be called in order in the text, and the caption of figure 7 is hidden. Fig 7 and 6 should be Fig 1 and 2 respectively. Alternatively, move the figure call later in the text or even better, merge one picture for QD and one for FP with Fig 1 (which will then contain 4 panels)

>>>Response: Thank you for bringing this to our attention. This error has been corrected to ensure proper naming order is applied consistently throughout the manuscript and have added a second quantum dot image figure (figure 2 in revised ms) in order to demonstrate the visual differences between the quantum dot colors

28) Figures formatting would benefit from a better standardization (e.g. font size and layout consistency. Figure captions lack adequate detail, making them less standalone. 

>>>RESPONSE: Thank you for bringing this to our attention. The figures have been revised to ensure consistency  across the manuscript, and additional details have been added to each figure caption to make them more self-explanatory and standalone.

Reviewer 2 Report

Comments and Suggestions for Authors

General comments

This manuscript evaluates the potential of quantum dots (QDs) as external markers for Medflies, which could aid in tracking insects in the field and support pest management through the sterile insect technique. The results indicate that QDs were less efficient compared to traditional fluorescent powder markers. The study rationale and objectives are clear, but the methods and results require clarification in many points. Figures can be improved for better quality and visual appeal. Additionally, I recommend reviewing the text for spelling and grammatical errors.

Below are my specific comments:

Abstract
Since QDs in this study did not exhibit the properties of an effective marker in most tests and no experiments in this work showed their versatility, the sentence, “These findings indicate that while QDs possess potential as marking agents, particularly given their versatility, specificity and colour range…” (lines 35-37), might mislead readers. I suggest rephrasing this sentence.

Introduction
I have no problem with this section. However, the language should be rechecked as I found some grammatical errors.

Materials and Methods

  1. Please specify the proportions of each ingredient used to prepare the artificial diet for insect rearing.
  2. Include information on the particle size and chemical composition of the QDs, and the particle size of fluorescent powder (if possible), as these characteristics could influence the results and help in interpretation.
  3. Lines 155-156 mention “All experiments used a concentration of 0.04 grams of fluorescent powder per 1,000 Medfly pupae.” However, in the optimal marking experiment, the ratio used was 0.0008 g per 50 pupae. I have tried to calculate. It appeared that it is equivalent to 0.016 g per 1,000 pupae. Please clarify this discrepancy.
  4. The first figure mentioned in the manuscript is Figure 7 (page 4), while Figure 1 appears later (page 6). This sequencing error must be corrected.
  5. The use of 7.5 mg/mL QDs (160 µL per 1,000 pupae) differs from the fluorescent powder quantity. If my calculations are correct, this corresponds to 1.2 mg (0.0012 g) of QDs per 1,000 pupae, which is lower than the fluorescent powder quantity used for marking. This issue should be concerned as it causes difficulty in comparison between QDs and fluorescent powder. Could the inferior performance of QDs be due to insufficient amounts?
  6. In section “2.4 Optimal marking technique,” only three conditions were tested without varying the marker quantity. Consider using the term “Suitable marking technique” instead.
  7. In section “2.6 Field flight assay and recapture rates,” please specify whether QDs and fluorescent powder markings were conducted simultaneously. If so, the selected particle colors (yellow, red, orange, and pink) are in similar tones. I am concerned that it might be difficult to differentiation between each color. To address this concern, include images of Medflies marked with all colors used in the study. Also, clarify which colors of QDs and fluorescent powder are shown in Figures 6 and 7.

Results

  1. In Figure 2, what do “N” and “Y” stand for? If they mean “No” and “Yes,” please use the full words. Additionally, presenting only column Y is enough, and Figures 2A, B, and C can be combined into a single figure.
  2. For the field flight assay, the traps were monitored at 24, 48, and 72 hours. Please specify which time point corresponds to the results in lines 342-343.
  3. According to the methods, only 1,000 pupae were used in each trial and treatment group (line 201). Please clarify how the total number of flies in Figure 3 exceeds 1,000 pupae.
  4. For the transference and stability experiment, only statistical results are shown. Please include the transference rates and clarify the statement in line 381, “a higher rate of marker loss over time,” with supporting data.
  5. The “Storage ability” experiment examined marked flies at 2, 4, and 8 weeks, but the results for individual time points are not shown. Figure 5 should indicate which time points the values represent.

Discussion

  1. The discussion includes various explanations but cites only 1–2 references per paragraph. I recommend adding more references to support these explanations.
  2. The authors did not discuss why QDs failed to adhere to the fly surface or why fluorescent powder performed better. This is a critical point to discuss, as it reflects the efficiency of QDs as markers.
  3. Lines 488-492: I don’t understand about the connection between non-flyers and self-grooming behavior. Please clarify this relationship and provide supporting references.

Author Response

Thank you for your thorough and insightful review of our manuscript. We appreciate the time and effort the referees have taken to provide detailed and constructive feedback. In the following response, we address each of your comments systematically.. Where possible, we have provided additional discussion points to address the major and minor concerns. We believe the revisions have strengthened the manuscript and improved its contribution to the field. Below, we detail the specific changes made in response to each of the comments raised.

We hope you find this acceptable and look forward to hearing from you.

Sincerely,

Dillon, on behalf of all authors

Abstract

  1. Since QDs in this study did not exhibit the properties of an effective marker in most tests and no experiments in this work showed their versatility, the sentence, “These findings indicate that while QDs possess potential as marking agents, particularly given their versatility, specificity and colour range…” (lines 35-37), might mislead readers. I suggest rephrasing this sentence.

>>>RESPONSE: The sentence originally starting at Line 36 in the original manuscript ("QDs possess potential as marking agents, particularly given their versatility, specificity, and colour range [...]") has been removed in the revised version. The mention of versatility, which was in reference to the various applications of QDs that are already being used, has also been removed. However, we have expanded and clarified some of the removed abstract points from the original manuscript in the revised introduction. This includes the addition of Lines 151–153 in the revised manuscript and the rewording of Line 122-123 in the original manuscript (now Lines 155–156 in the revised manuscript) for improved clarity on what we were referring to as specificity. The discussion on colour range and its advantages has also been expanded in the revised manuscript in Lines 136–144, which should help clarify why the color range and other advantages are important for QDs.

Introduction

  • I have no problem with this section. However, the language should be rechecked as I found some grammatical errors.

>>>RESPONSE: Thank you for bringing this to our attention, the grammar has hopefully now been corrected in the revised version

Materials and Methods

1) Please specify the proportions of each ingredient used to prepare the artificial diet for insect rearing.

>>>RESPONSE: Thank you for raising this point, as it is an important consideration for replicating the experiment. However, since the pupae were sourced from Fruit Fly Africa, we requested information on the specific proportions they use. Unfortunately, management deemed this information too sensitive to disclose. For the adults they are given sugar and water ad libitum separately which has been clarified in the methodology (line 292-293 revised ms; line 154 in original ms)

2) Include information on the particle size and chemical composition of the QDs, and the particle size of fluorescent powder (if possible), as these characteristics could influence the results and help in interpretation.

>>>RESPONSE: Thank you for this point and we agree. We have now updated the methodology section to now include all the necessary information on the QDs (line 313-320 in revised ms; line 165 in original ms)

3) Lines 155-156 mention “All experiments used a concentration of 0.04 grams of fluorescent powder per 1,000 Medfly pupae.” However, in the optimal marking experiment, the ratio used was 0.0008 g per 50 pupae. I have tried to calculate. It appeared that it is equivalent to 0.016 g per 1,000 pupae. Please clarify this discrepancy.

>>>RESPONSE: Thank you for bringing this discrepancy to our attention. We have revised the manuscript to clarify the differences in fluorescent powder concentrations used in the experiments (lines 303-305 in the revised manuscript) and have added an explanation in the ‘suitable marking technique’ section (lines 431-438 in the revised manuscript). The variation in concentration was due to adjustments made for the suitable marking technique experiment, where a smaller sample size required a lower concentration to prevent over-marking. Additionally, working with extremely small quantities was impractical with the available equipment, and the original concentration was optimized for large-scale applications. However, as the objective of the experiment was to determine the most suitable marking technique rather than standardize specific dosages, we deemed these adjustments did not affect the overall findings.

4)The first figure mentioned in the manuscript is Figure 7 (page 4), while Figure 1 appears later (page 6). This sequencing error must be corrected.

>>>RESPONSE: The sequencing has been fixed.

5)The use of 7.5 mg/mL QDs (160 µL per 1,000 pupae) differs from the fluorescent powder quantity. If my calculations are correct, this corresponds to 1.2 mg (0.0012 g) of QDs per 1,000 pupae, which is lower than the fluorescent powder quantity used for marking. This issue should be concerned as it causes difficulty in comparison between QDs and fluorescent powder. Could the inferior performance of QDs be due to insufficient amounts?

>>>RESPONSE: We never intended to apply the two sets of markers at equal concentrations since they are fundamentally different marking approaches. Theoretically one could apply a single QD and if it remains clearly attached would be sufficient to mark the fly.

6) In section “2.4 Optimal marking technique,” only three conditions were tested without varying the marker quantity. Consider using the term “Suitable marking technique” instead.

>>>RESPONSE: We agree with the renaming and this is reflected in the revised manuscript.

7) In section “2.6 Field flight assay and recapture rates,” please specify whether QDs and fluorescent powder markings were conducted simultaneously. If so, the selected particle colors (yellow, red, orange, and pink) are in similar tones. I am concerned that it might be difficult to differentiation between each color. To address this concern, include images of Medflies marked with all colors used in the study. Also, clarify which colors of QDs and fluorescent powder are shown in Figures 6 and 7.

>>>RESPONSE: For Section 2.6: Field Flight Assay and Recapture Rates, we have clarified in line 494-496 of the revised manuscript that QD and fluorescent powder markings were conducted simultaneously. We do not believe that color differentiation was an issue, as QDs exhibit a narrow emission spectrum, which allows for clear distinction between different colors. This has been further explained in the introduction (lines 133-144)  in the revised manuscript). Additionally, we have included an image of all QDs side by side in the revised manuscript to visually demonstrate the distinct color differences (Figure 2)

Results

1) In Figure 2, what do “N” and “Y” stand for? If they mean “No” and “Yes,” please use the full words. Additionally, presenting only column Y is enough, and Figures 2A, B, and C can be combined into a single figure.

>>>RESPONSE: Thank you for the helpful suggestion. Figure 4 (previously Figure 2 in the original ms) has now been combined into a single figure.

2) For the field flight assay, the traps were monitored at 24, 48, and 72 hours. Please specify which time point corresponds to the results in lines 342-343.

>>>RESPONSE: The results reported on lines 342–343 in the original manuscript referred to all time points combined. This has now been clarified more explicitly in the revised manuscript (line 825) to ensure accurate interpretation of the data.

3) According to the methods, only 1,000 pupae were used in each trial and treatment group (line 201). Please clarify how the total number of flies in Figure 3 exceeds 1,000 pupae.

>>>RESPONSE: Thank you for bringing this to our attention. Figure 3 represents the total across all three replicates (1,000 pupae × 3 replicates), which is why the graph stops just before 3,000 for each color. This has now been more clearly stated in the caption of Figure 3 to minimise any confusion

4) For the transference and stability experiment, only statistical results are shown. Please include the transference rates and clarify the statement in line 381, “a higher rate of marker loss over time,” with supporting data.

>>>RESPONSE:. In the revised manuscript, the transference rates are now explicitly stated in lines 979-985, and the statement in line 381 of the original manuscript, “a higher rate of marker loss over time,” now has supporting data provided in lines 989 – 999 to help clarify the findings.

5) The “Storage ability” experiment examined marked flies at 2, 4, and 8 weeks, but the results for individual time points are not shown. Figure 5 should indicate which time points the values represent.

>>>RESPONSE: Figure 5 (now Figure 7 in the revised manuscript) has been replaced with a graph that includes the different time points, as we believe this better represents our results and improves clarity in data interpretation.

Discussion

1) The discussion includes various explanations but cites only 1–2 references per paragraph. I recommend adding more references to support these explanations.

>>>RESPONSE: More references have been added as suggested. Please let us know if more is needed.

2) The authors did not discuss why QDs failed to adhere to the fly surface or why fluorescent powder performed better. This is a critical point to discuss, as it reflects the efficiency of QDs as markers.

>>>RESPONSE: We have added an extra paragraph in the discussion to further clarify key aspects, including why QDs failed to adhere effectively. This additional explanation can be found in the revised manuscript (lines 1346–1379) which should provide more clarity to why they may not have worked as well as anticipated.

3) Lines 488-492: I don’t understand the connection between non-flyers and self-grooming behavior. Please clarify this relationship and provide supporting references.

>>>RESPONSE: The relationship between non-flyers and self-grooming behavior has now been clarified in the revised manuscript (lines 1233-1242 previously lines 492-496 in the original ms). Tephritid flies are well-documented for their persistent grooming behavior, which results in the gradual removal of superficial dye particles from their bodies (Schroeder & Mitchell, 1981; Enkerlin et al., 1996). In our study, the higher proportion of non-flyers observed in the fluorescent powder group may be attributed to this increased grooming activity in response to the marking process, hence why there was a lower proportion of flyers. We acknowledge that while this is a plausible explanation for our findings, further research is required to fully establish the relationship between fluorescent powder markings, self-grooming behavior, and flight performance in Medflies.

Round 2

Reviewer 1 Report

Comments and Suggestions for Authors

I thank the authors for addressing my concerns to the best they could within the limited allotted time. I do not have any more suggestions that would improve the paper at this stage.